# BIOPOINT: A particle-based model for probing nuclear mechanics and cell-ECM interactions via experimentally derived parameters

Sandipan Chattaraj*, Julius Zimmermann(iD), Francesco Silvio Pasqualini(iD)*

Dept of Civil Engineering and Architecture, University of Pavia: Universita degli Studi di Pavia, Pavia, ITALY

* sandi.chattaraj@gmail.com (SC); francesco.pasqualini@unipv.it (FSP)

## Abstract

Morphogenesis arises from biochemical and biomechanical interactions across multiple spatial and temporal scales. Experimental studies alone cannot fully resolve these dynamics, motivating computational models. Subcellular element modeling (SEM) is well suited for simulating emergent cellular and tissue morphologies, but traditional SEM frameworks do not explicitly include nuclear deformation or direct cell–extracellular matrix (ECM) interactions: capabilities typically associated with continuum approaches based on the finite-element method (FEM) approaches. FEM excels at modeling cell and tissue mechanics, but struggle to accommodate the large, non-linear deformations driven by local, geometry-changing events that define morphogenesis. Here, we introduce BIOPOINT, a particle-based framework that augments SEM with FEM-like mechanical capabilities by incorporating (1) a deformable, multi-particle nucleus capable of capturing nuclear stress and strain distributions and (2) an explicit ECM layer represented by structured static particles with tunable adhesive potentials. To ensure biological relevance, we calibrate BIOPOINT against single-cell indentation experiments (SKOV3). We then apply this calibrated parameter set, without additional refitting, to two independent scenarios: (i) cell (EC and hMSC) spreading on ECM micropatterns, capturing qualitative coupling between cell and nuclear shape; and (ii) confined migration (MDA-MB-231) through rigid constrictions, qualitatively reproducing the characteristic sequence of nuclear elongation and partial recovery. Differently from previous work, we present a full SEM model that uses heterogeneous particles to model nuclei, cells, and ECM via phase separation. By combining SEM's strength in modeling emergent cell and tissue geometry with a mechanically sound handling of nuclear and ECM interactions, BIOPOINT provides a versatile platform for studying cell behaviors, like shape acquisition and migration through confinement that are relevant to morphogenesis. Implemented within the widely used, open-source LAMMPS ecosystem, BIOPOINT offers an accessible and extensible tool for the community.

**Data availability statement:** The data that supports the findings of this study are available on Zenodo (https://doi.org/10.5281/zenodo.15223446) and the code is maintained on GitHub: https://github.com/Synthetic-Physiology-Lab/biopoint.

**Funding:** This work was supported by the European Research Council (ERC) Starting Grant #852560 to FSP. SC received his salary from the ERC Starting Grant #852560. The funders had no role in study design, data collection and analysis, decision to publish, or preparation of the manuscript.

**Competing interests:** The authors have declared that no competing interests exist.

## Author summary

Morphogenesis is the process by which a living thing grows from a single cell to a full organism. Investigating the mechanical forces at play during this process is important for disease modeling and synthetic development of organoids. Computational studies of morphogenesis require accounting for the pushing and pulling forces through which complex shapes of cells and tissues emerge. Subcellular element modeling (SEM) is a computational technique which can capture emergent shapes while also providing information about subcellular mechanics. This work aims to implement full SEM, where the entire cell is assumed to be made of particles, while bringing to the table the following important features: 1. a deformable nucleus made of multiple particles, 2. cell-ECM (extracellular matrix) interactions through explicit ECM particles and 3. calibration of model parameters directly from experiments on a cell type and qualitatively validating them on different cell types. We determined the elastic stiffness and viscosities of the cell and nucleus from the force-time data of single cell AFM experiments. With the calibrated parameter values, we can predict the evolution of cell and nuclear shapes in cell spreading experiments on various ECM geometries and in cells migrating through narrow constrictions.

## Introduction

Morphogenesis is the process by which tissues and organs develop their shape and function, both in vivo during development [1] and in vitro in organoids [2–4] or organs-on-chips [5,6]. This process is controlled by complex biochemical and biomechanical interactions across multiple spatial and temporal scales. Understanding these interactions is crucial for deciphering developmental processes, modeling disease progression, and designing biomimetic engineered tissues. While experimental approaches have provided significant insights, they are often constrained by spatial resolution and temporal control, making it challenging to capture the emergent mechanical behavior of cells and tissues in physiologically relevant contexts [1,7]. One fundamental challenge in morphogenesis is the mechanical interplay between cells, their internal organelles, and their external microenvironment. For example, the nucleus is the largest organelle in the cell and plays a key role in regulating cell migration, differentiation, and mechanotransduction [8,9]. Likewise, the extracellular matrix (ECM) provides biochemical and mechanical cues that guide cellular behavior and emergent tissue structures [10, 11]. Therefore, the ability to quantitatively model nuclear mechanics and cell-ECM interactions is critical for understanding how mechanical forces drive morphogenesis.

 Computational models provide a powerful means of integrating experimental data and explaining emergent behaviors [12]. While classical FEM approaches have provided detailed nuclear stress-strain predictions, they typically struggle with large cell deformations and tissue heterogeneity [13,14]. Vertex and Cellular Potts

Models (CPM) can simulate large deformation and local events but struggle with representing distributed cell and nuclear mechanics [15]. Subcellular Element Modeling (SEM) has emerged as a promising framework for simulating multicellular and subcellular mechanics [16–18]. SEM enables the modeling of emergent shapes by treating cells as ensembles of coarse-grained particles interacting via empirically defined potentials. Recently, we introduced SEMM or SEM$^2$, which allows the calculation of multiscale mechanics in SEM simulations by assessing stress and strain at the particle, cell, and tissue levels [19]. However, a key limitation of existing SEM models is their inability to capture nuclear mechanics and cell-ECM interactions explicitly. Prior SEM implementations have typically represented the nucleus as a single rigid particle [18,19], which fails to account for nuclear deformability and the role of intracellular forces in nuclear positioning and chromatin remodeling. Similarly, the ECM has generally been modeled as an implicit solvent rather than a discrete, separate environment directly interacting with cells [18,19]. Finally, SEM parameters were traditionally derived by homogenizing bulk elastic stiffness and viscosity of non-adherent cells [16,18,19], leading to model parameters that might not be relevant for cell-ECM interactions during morphogenesis. As a result, current SEM frameworks cannot accurately model processes such as nuclear deformation during confined migration or how ECM geometry influences cell spreading and mechanosensing. By contrast, continuum approaches and the finite element methods (FEM) can reconstruct nuclear stress–strain fields and cell–ECM coupling at the single-cell level (e.g., Estabrook et al., [20]; Maxian et al., [21]), but they rely on prescribed meshes and boundary conditions and are not easily embedded into large-scale morphogenetic simulations with large deformations, neighbor exchanges, and topological changes. Our goal here is therefore not to replace FEM, but to extend SEM so that these same classes of nuclear and cell–ECM mechanics can be addressed in a discrete, particle-based framework where cell and nuclear shape emerge from local interactions.

Discrete models naturally capture biological processes and are effective in resolving cell mechanics at the cell membrane as a balance of forces [22]. Energy is related to force via integration and interconversions within the Cellular Potts formalism [23] or vertex models [24]. However, the cell mechanics in these cases is limited to cell membranes [7,25]. Yet, the mechanical forces at play in the intracellular space, or cytoplasm, can influence the mechanical behavior of entire tissues [26,27]. Therefore, SEM remains the discrete modeling framework with the advantage of being able to model cell and tissue shapes as emergent properties of subcellular interactions [28,29]. Yet, it lacks the ability to handle the cell nucleus and the ECM as heterogeneous particle types. Previous discrete approaches have modeled a deformable nucleus using particles coupled to lattice-based descriptions of the cell and ECM in custom codes (e.g., Nematbakhsh et al., [28]), underscoring the demand for such capabilities but leaving a gap for a fully SEM-based, open-source implementation that treats the cell, nucleus, and ECM within a single particle framework. To address these limitations, we propose two advancements. First, we introduce a deformable nucleus, represented by phase-separated interacting particles, enabling simulation of nuclear strain and stress distributions. Second, in order to recapitulate in vitro experiments, we explicitly represent the ECM as a planar arrangement of static particles interacting with the cell via tunable adhesive potentials. Since everything in the simulation is modeled explicitly via discrete particles, our simulations resemble paintings in the style of famed pointillists Georges Seurat (1859–1891) and Paul Signac (1863–1935). In their honor, we called this framework BIOPOINT. Notably, BIOPOINT enables quantitative predictions of nuclear mechanics and cell-ECM interactions, thanks to the explicit representation of nuclear and ECM particles. Further, the parameters governing potentials can be calibrated directly against experimental data collected in relevant biological conditions, including adherent cells on various ECM environments. To showcase BIOPOINT, we followed a three-step approach. First, we calibrated model parameters using single-cell indentation experiments, ensuring that nuclear and cytoplasmic mechanical properties are quantitatively aligned with experimental force-time data [30,31]. Second, we validated these parameters through cell-spreading simulations on 2D ECM micropatterns, assessing whether BIOPOINT can reproduce experimentally observed relationships between ECM geometry and nuclear shape dynamics [32,33] that were not used for parameter fitting. Finally, we applied the fitted model to the simulations of nuclear deformation during 3D confined migration, testing whether BIOPOINT can replicate experimentally reported nuclear strain and shape changes as cells traverse narrow

constrictions [34,35]. This approach aims to demonstrate that BIOPOINT extends to the nucleus and the ECM the ability of SEM to simulate emergent shape and mechanical changes, filling a gap in morphogenesis modeling. In doing so, BIO-POINT complements established continuum frameworks by making experimentally calibrated nuclear and ECM mechanics available within a community-maintained LAMMPS implementation, lowering the barrier to deploying SEM in realistic morphogenetic simulations rather than merely offering an alternative numerical discretization.

## Results

### Theory of SEM

Simulating cells and tissues with SEM implies solving the Langevin equation for an arbitrary number of subcellular particles ($N_p$) [16].

$$\eta \dot{y}_{i,j,k} = \xi_{i,j,k} + F_C(y_{i,j,k})$$

(1)

In Eq. (1), $\eta$ is the viscous drag coefficient and $y_{i,j,k}$, $\dot{y}_{i,j,k}$ are the position and velocity of each particle. We chose the indexes $i$, $j$, $k$ to denote particle number, particle type, and cell-id, respectively. With this notation, $\xi_{i,j,k}$ is the thermal fluctuation felt by each particle and $F_C(y_{i,j,k})$ is the net pairwise force acting on it. In particular, Newman used the following Morse-like elastic potential ($V(d)$ where $d$ is the inter-particle distance) to enforce adhesion and volume exclusion (Fig 1B and Eq. 2) [16,18].

$$V(d) = u_0 e^{2\rho\left(1-\frac{d^2}{d_{eq}^2}\right)} - \alpha u_0 e^{\rho\left(1-\frac{d^2}{d_{eq}^2}\right)}$$

(2)

In Eq. (2), $u_0$ is the potential's well depth, $\rho$ and $\alpha$ are scaling and shifting factors, and $d_{eq}$ is the equilibrium distance between particles. To limit the computational cost, these potentials focus on short-range interactions and cut off at 2.5 times the equilibrium distance. To assign the forces in Eq. (1) based on cell-level rheology, Newman assumed $N_p$ densely packed particles connected by springs and recovered the following relationships [16]

$$d_{eq} = 2R_{cell}\left(\frac{p_d}{N_p}\right)^{\frac{1}{3}}$$

(3a)

$$\eta = \frac{\eta_0}{N_p}$$

(3b)

$$\kappa = \kappa_0 N_p^{-\frac{1}{3}}\left(1 - \lambda N_p^{-\frac{1}{3}}\right)$$

(3c)

$$u_0 = \kappa \frac{d_{eq}^2}{(8\rho^2)}$$

(3d)

In Eqs. (3a-3d), $p_d$ is the sphere close packing density, $R_{cell}$, is the radius of the cell, $\kappa_0$ and $\eta_0$ are the stiffness and viscosity of the cytoplasm, respectively. The viscous force acting on each particle is given in the left-hand side (LHS) of Eq. (1). $\lambda$ is a tuning coefficient which is an adjustment to take care of the departure from a cubic lattice arrangement used to derive the elastic modulus of the network [16]. The list of parameters and their values are listed in Table 1. We have inherited some of the basic parameter values, related to the functional form of the potential, such as the scaling factor, shifting

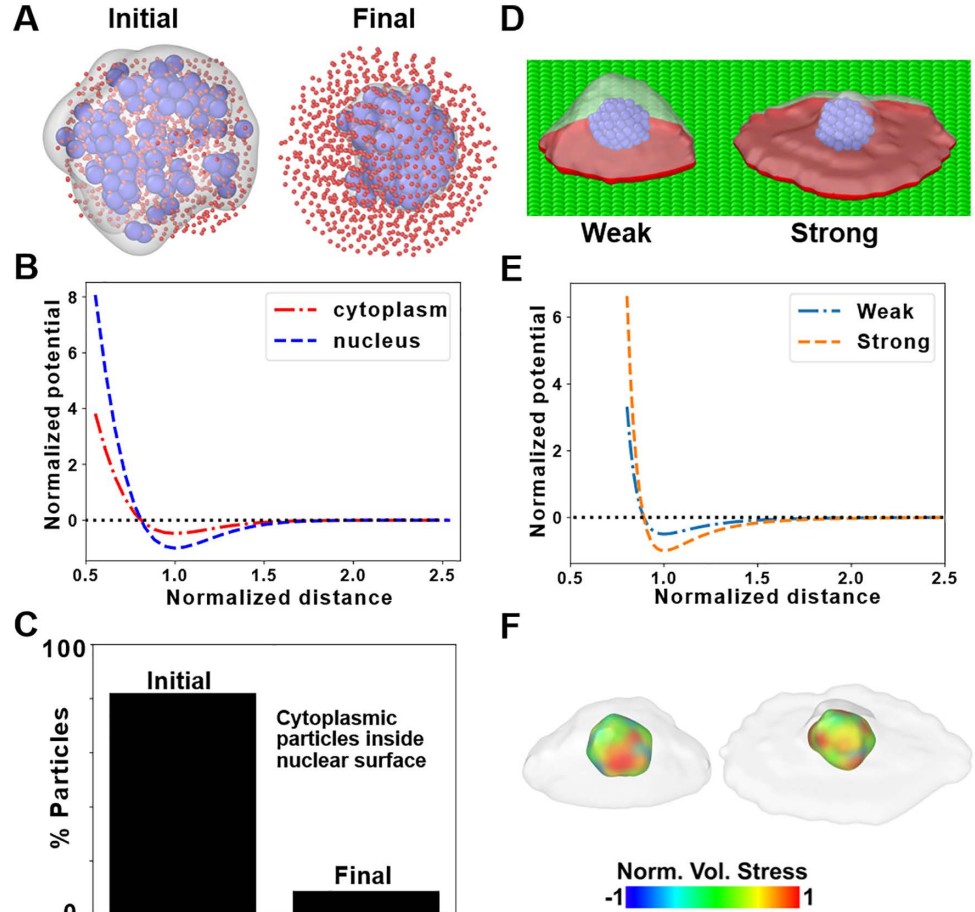

**Fig 1. BIOPOINT: Extending SEM² to incorporate a multi-particle nucleus and new particle type for ECM. (A)** Representation of the initial and phase-separated structure of an *in silico* cell with Np = 1000, where 900 cytoplasmic (red) and 100 nuclear (blue) particles have been created randomly within a cylindrical space. To visualize the nucleus, we create a surface rendering of nuclear particles. Though the sizes of cytoplasmic and nuclear particles are the same, the former is shown as smaller than the latter for visualization purposes. **B)** Plot of potential versus distance between cytoplasmic particles and between nuclear particles. **C)** Percentage of cytoplasmic particles within the nuclear surface in the initial and final samples, with phase-separated nuclear particles. **D)** Weak and strong interaction with extracellular matrix (ECM) particles can lead to different degrees of cell spreading. Projected surface area at the bottom is shown in red. **E)** Plot of normalized potential versus distance between ECM and cytoplasmic particles for the weak and strong interactions. **F)** Spatial distribution of stresses in the nucleus for the weakly and strongly adhering cells, shown in **D**. For visualization purposes, surface meshes have been generated on top of the respective particles, representing the cell membrane (in white) and the nuclear membrane (color-coded according to the per-particle stress). The visualizations of all cells, nuclei and ECM such as in A, D, F, have been generated using the visualization software, OVITO [40].

factor, tuning coefficient from an earlier SEM model [18]. The model is calibrated on a particular cell type and ideally it should be applied on that same cell type and similar experimental conditions for quantitative match with experiments.

## BIOPOINT uses heterogeneous discrete particle types to model the cell nucleus and ECM.

SEM traditionally treated the nucleus as a single, rigid particle within the cytoplasm, preventing accurate simulation of nuclear deformability and intracellular stress distributions [16,18,19]. Likewise, classical SEM frameworks have implicitly modeled the extracellular matrix (ECM), lacking the ability to capture the influence of ECM geometry on cell spreading and migration. These limitations restrict the ability to model key morphogenetic processes, including nuclear deformation

**Table 1. Parameters used in SEM2.**

| Parameter | Description | Value |
|---|---|---|
| $R_{cell}$ | Cell radius | 10 μm |
| $m_{cell}$ | Cell mass | 3.1 ng |
| $N_p$ | Total number of particles in a cell | 1000 |
| $p_d$ | Sphere close packing density | 0.74 |
| $\rho$ | Scaling factor | 2 |
| $\alpha$ | Shifting factor | 2 |
| $\kappa_0$ | Cell stiffness | $1.2 \times 10^{-2}$ N m$^{-1}$ |
| $\eta_0$ | Cell viscosity | $5 \times 10^{-2}$ N s m$^{-1}$ |
| $\lambda$ | Tuning coefficient | 0.75 |
| $m_p$ | Mass of each particle | $3.1 \times 10^{-3}$ ng |

during confined migration and ECM-guided mechanosensing. To overcome these issues, BIOPOINT introduces a deformable nucleus and an explicit ECM representation, as described below.

We first generated a multiparticle nucleus phase separated from the cytoplasmic particles. We initialized a single cell in the simulation as an ensemble of 1000 interacting particles, of which 100 (10%) were designated as nuclear particles (Fig 1A). These particles were assigned a stronger self-interaction potential (nuclear-nuclear adhesion) than cytoplasmic particles to induce phase separation. If $\kappa_{nuc}$ and $\eta_{nuc}$ are respectively the stiffness and viscosity of the nucleus, then the relative stiffness of cytoplasmic-nuclear interaction, in comparison to cytoplasmic stiffness is represented by the parameter, $k_{12}$.

$$k_{12} = \frac{u_0(1,2)}{u_0(1,1)}$$

(4)

In Eq. 4, $u_0(1,2)$ and $u_0(1,1)$, represent the potential well depths for the cytoplasm–nucleus (types 1,2) and cytoplasm–cytoplasm (types 1,1) interactions, respectively. Eq. (3d) generalizes to interactions between particle types p and q as $u_{0(p,q)} = \kappa_{(p,q)} \frac{d_{eq(p,q)}^2}{(8\rho^2)}$, where $d_{eq(p,q)}$ is set by the equilibrium spacing of the two particle radii (e.g., $d_{eq(p,q)} = R_p + R_q$). In BIO-POINT, cytoplasmic and nuclear particles are equal-sized, so $d_{eq(1,2)} = d_{eq(1,1)}$, and we set $\frac{u_0(1,2)}{u_0(1,1)} = \frac{\kappa_{(1,2)}}{\kappa_{(1,1)}} = k_{12}$. The nuclear particles were assigned a higher stiffness potential ($\kappa_{nuc} = 2\times \kappa_0$) to reflect the experimentally observed higher stiffness of the nucleus compared to the surrounding cytoplasm [36] (Fig 1B). We then performed an equilibration step, allowing the nucleus to coalesce within the cytoplasmic particle ensemble. After equilibration, the nuclear particles successfully segregated into a distinct phase within the cytoplasm, forming a cohesive, deformable nucleus (Fig 1C). Nuclear and cytoplasmic particles were intermixed in the initial random configuration. Still, after phase separation, fewer than 8% of cytoplasmic particles remained inside the nuclear boundary (Fig 1C). Considering that we have started from a completely mixed configuration of equal-sized particles, the potentials of the two types of particles have the same functional form and there is no nuclear membrane, it is exponentially more difficult from a computational standpoint to obtain a better phase separation [37]. The nuclear stiffness parameter ($\kappa_{nuc}$) effectively maintained the nucleus as a mechanically distinct subregion, consistent with biological observations of nuclear mechanics [36].

To evaluate the explicit ECM representation, we constructed an ECM layer consisting of static particles arranged in a plane at the base of the simulation domain (Fig 1D). These ECM particles exerted tunable adhesive interactions with cytoplasmic particles via a Lennard-Jones (LJ) potential. The attraction between cytoplasmic and ECM particles is modeled by a Lennard-Jones (12–6) potential:

 

$$V(d) = 4\varepsilon \left[ \left(\frac{\sigma}{d}\right)^{12} - \left(\frac{\sigma}{d}\right)^{6} \right] \quad d < d_c$$

(5)

In Eq. 5, $d$ is the distance between the cytoplasmic and ECM particles, $\sigma$ is the zero-crossing distance, $\varepsilon$ is the potential well-depth, and $d_c$ is the cutoff distance. Details about the determination of LJ potential parameters is provided in Table A in S1 Appendix. Table 2 lists the additional parameters needed in BIOPOINT to address multi-particle nucleus and cell-ECM interactions. We then systematically varied the cell-ECM adhesion strength ($\varepsilon$) to test how this parameter influences cell spreading behavior. Cells on the ECM substrate exhibited distinct spreading behaviors depending on the adhesion potential. When the cell-ECM adhesion strength was low, cells remained compact and retained a near-spherical shape (Fig 1D, left). Increasing the adhesion strength resulted in greater cell spreading (Fig 1D, right), with the projected area being more for a higher value of $\varepsilon$, which corresponds to well depth of the LJ potential (Fig 1E, Fig A in S1 Appendix). Notably, high adhesion levels led to maximal cell spreading, consistent with experimental studies on ECM-guided migration [38,39]. The spatial distribution of stress in the nucleus for the cells with different degrees of spreading can be obtained from per-particle stress computation and visualized (Fig 1F).

These results confirm that BIOPOINT integrates nuclear deformability and explicit ECM interactions into a SEM framework. Forming a phase-separated nucleus demonstrates that our approach can model intracellular compartmentalization and stress-strain distributions at the nuclear level. Additionally, the planar ECM representation allows for the systematic tuning of adhesion forces, enabling the study of ECM-guided cell behavior. These enhancements overcome key limitations of previous SEM-based models and provide a biophysically relevant platform for studying nuclear mechanics and cell-ECM interactions.

## Calibration of model parameters via single-cell indentation experiments

Previous SEM attempts lacked direct experimental calibration, limiting their accuracy in predicting cellular and nuclear mechanics [16,18,19]. To ensure BIOPOINT accurately captures these properties, we calibrated its parameters using single-cell indentation experiments, a standard technique for measuring cellular viscoelasticity and nuclear stiffness [30,31]. By simulating Atomic Force Microscopy (AFM) indentation and comparing force-time responses with experimental data, we aimed to identify key mechanical parameters using Uncertainty Quantification (UQ) and determine the elastic and viscous properties of the nucleus and cytoplasm using non-linear optimization.

We designed an *in silico* indentation experiment replicating the protocol of Hobson *et al*. [31] (Fig 2A). A BIOPOINT cell was spread on an ECM layer, with the nucleus positioned near the top surface. This is a result of the fact that there is interaction between the cytoplasmic and ECM particles but not between nuclear and ECM particles. To prevent excessive lateral spreading, peripheral cytoplasmic particles were immobilized, mimicking focal adhesions (Fig 2B). Indentation was

**Table 2. Additional parameters used in BIOPOINT.**

| Parameter | Description | Value |
|---|---|---|
| $N_n$ | Number of nuclear particles | 100 |
| $\kappa_{nuc}$ | Nuclear stiffness | $4.5 \times 10^{-2}$ N m$^{-1}$ |
| $\eta_{nuc}$ | Nuclear viscosity | $2.1 \times 10^{-1}$ N s m$^{-1}$ |
| $k_{12}$ | Relative strength of cytoplasmic-nuclear potential | 1.51 |
| $\sigma$ | Cell-ECM potential: zero-crossing distance | 5 µm |
| $\varepsilon$ | Cell-ECM potential: potential well-depth | 60000 zJ |
| $d_c$ | Cell-ECM potential: cutoff distance | 15 µm |

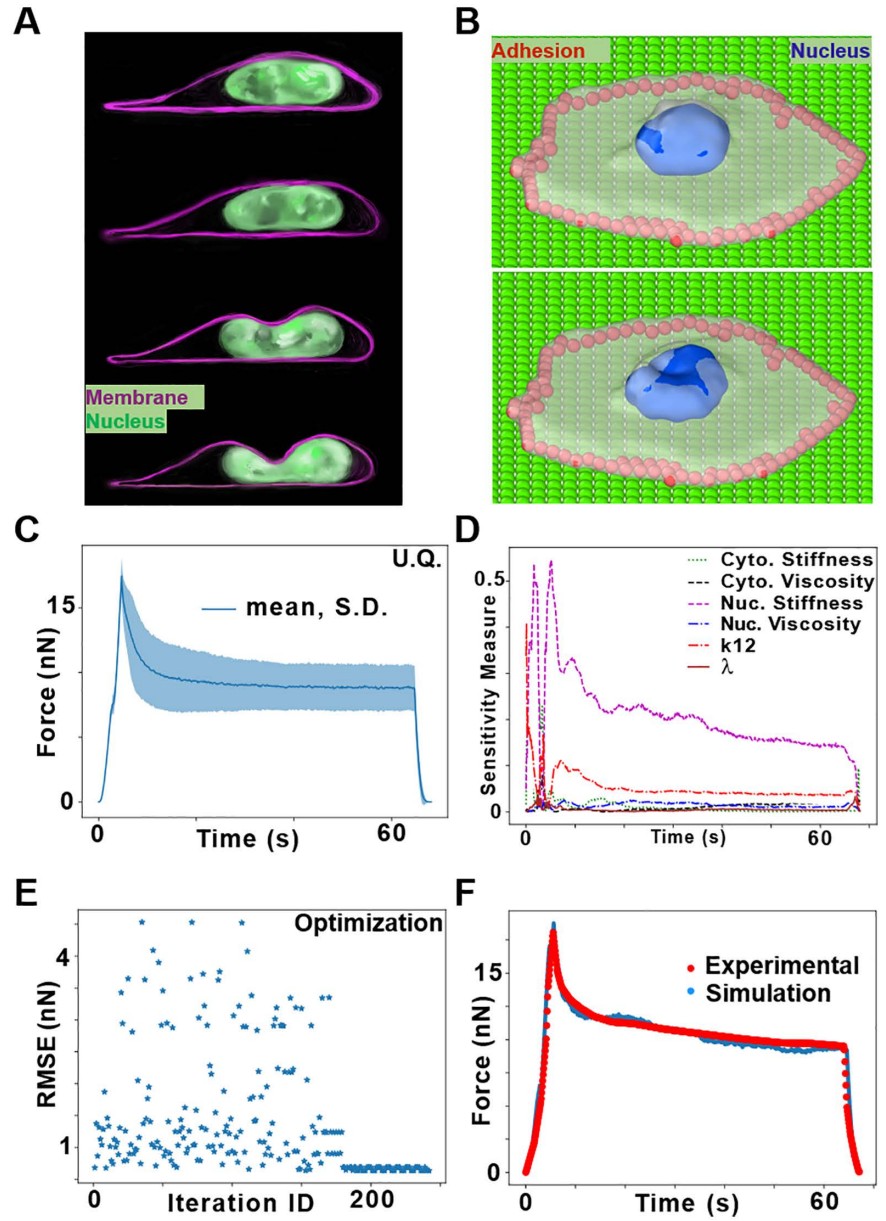

**Fig 2. Calibration of model parameters from the spherical indentation of a single cell. A)** Schematic of fluorescence image of SKOV3 cell undergoing indentation, inspired by Hobson *et al.* [31]. **B)** Representation of simulated cell, showing cell membrane (white), nucleus (blue), and peripheral constrained particles (red), before and after indentation. **C)** Uncertainty quantification (UQ) of parameters: Plot of mean and standard deviation of the force values versus time. Polynomial chaos expansion (PCE) $\mu \pm \sigma$ band of the indentation force vs time under Normal priors ($\sigma = 0.1 \cdot \mu$) centred at Table 3 nominal values for six parameters. **D)** First-order Sobol indices $S_i(t)$ from the same PCE, showing variance contributions of each parameter over time. (Higher $S_i \rightarrow$ larger share of output variance.) **E)** Optimization: Plot of RMSE values for each iteration during the optimization. **F)** Force-time plot of literature experiment and optimized simulation of single cell indentation.

performed using a rigid spherical tip moving in a displacement-controlled manner (S1 Video). In the loading phase, the tip compresses the cell and nucleus. During the hold phase, the tip remains stationary, allowing force relaxation. Finally, in the unloading phase, the tip retracts, relieving compression. Throughout the experiment, force exerted on the tip was

recorded, generating force-time curves comparable to AFM data. To simplify the analysis, we assumed no adhesion between the tip and the cell, avoiding artifacts observed in real AFM retraction phases. The spatial distribution of nuclear stress and strain can be computed and visualized during the entire simulation (S2 and S3 Videos).

The simulated force-time curve exhibited a three-phase response, with force increasing during loading, relaxing during the hold phase, and returning to baseline in unloading (Fig 2C). This response closely matched experimental AFM indentation data [31], confirming that BIOPOINT accurately captures single-cell compression mechanics. Then, we performed UQ to assess which parameters most influence indentation response. Nuclear stiffness and nuclear-cytoplasmic interaction potential ($k_{12}$) had the strongest effect on force-time curves (Fig 2D). The tuning coefficient ($\lambda$) had negligible influence, suggesting that this parameter doesn't contribute much to cellular and nuclear mechanics. Cytoplasmic viscosity and stiffness contributed but played a secondary role. These results confirm that nuclear mechanics drive indentation responses, reinforcing the need for explicit nuclear modeling in BIOPOINT.

To refine BIOPOINT's mechanical parameters, we performed non-linear optimization by minimizing the root mean square error (RMSE) between our simulation results and the experimental force-time curve. Upon convergence (Fig 2E), BIOPOINT learned parameter values that successfully matched experimental force-time curves, confirming that the model's elastic and viscous properties aligned with real cell behavior (Fig 2F). While we acknowledge that this might not be a unique set of parameter values which give us a good match with the experimental force-time curve, since we have probed a physiologically relevant parameter range and the initial guesses were obtained based on the physical meaning of parameter values, we assume that we determine properties which are close to reality. Thus, BIOPOINT reproduces the experimental force–time curve for the SKOV3 indentation dataset used for calibration, identifies dominant mechanical parameters, and yields an optimized parameter set for this fixed resolution ($N_p = 1000$). Importantly, the resolution of BIOPOINT—and thus the number of particles in a cell, $N_p$ — must be fixed prior to calibration. If $N_p$ is changed, particle size changes, and the nuclear phase separation should be redone. This requires re-equilibration and re-calibration before any quantitative comparison with experiments. We include Fig G in S1 Appendix as a qualitative illustration that changing $N_p$ can affect the indentation response.

## Qualitative comparison: Cell Spreading *on* ECM Patterns

Having calibrated BIOPOINT's mechanical parameters on SKOV3 indentation data (Fig 2), we next assessed the model's predictions for cell and nuclear shape evolution during ECM-guided spreading using the same parameter set without additional refitting. ECM geometry is a key regulator of cell mechanics, influencing morphogenesis, junction remodeling, and spatial organization within tissues [32,41]. We simulated cell spreading on micropatterned ECM substrates to achieve three goals. First, we assess whether BIOPOINT captures the qualitative ordering of cell shape evolution on different ECM geometries. Second, we quantify nuclear shape changes and compare qualitative NSI–CSI trends to published micropattern datasets (EC and hMSC; Fig 3G), noting these are different cell types than the calibration dataset. Third, we isolate key parameters governing nuclear deformation.

We modeled a single cell spreading on ECM micropatterns with distinct geometries: a circle (Fig 3A), a square (Fig 3B), a triangle (Fig 3C), and a series of elongated rectangles (Fig 3D, S4 Video). The ECM was represented as a layer of immobile particles that interacted with the cytoplasmic particles via adhesive potentials. Surrounding inert particles mimicked non-adhesive substrates (such as the glass on an experimental coverslip), resulting in a confined environment where cells can adhere only to the intended ECM shape. The stress distribution in the spread-out cell can be visualized from the computation of per-particle stress (Fig B in S1 Appendix). Please note that we haven't explicitly modelled membrane particles. However, as an emergent property of the simulations, we find that there is a greater presence of positive/tensile stress on the peripheral particles and negative/compressive stress on the particles inside the cell. Cell spreading was quantified over time using the Cell Shape Index (CSI) and Nuclear Shape Index (NSI), computed as: $CSI = \frac{4\pi A}{P^2}$ where A is the cell area and P is the perimeter. CSI values near 1 indicate a circular shape, while lower values reflect elongation (Fig 3E). NSI was measured similarly to track nuclear deformation.

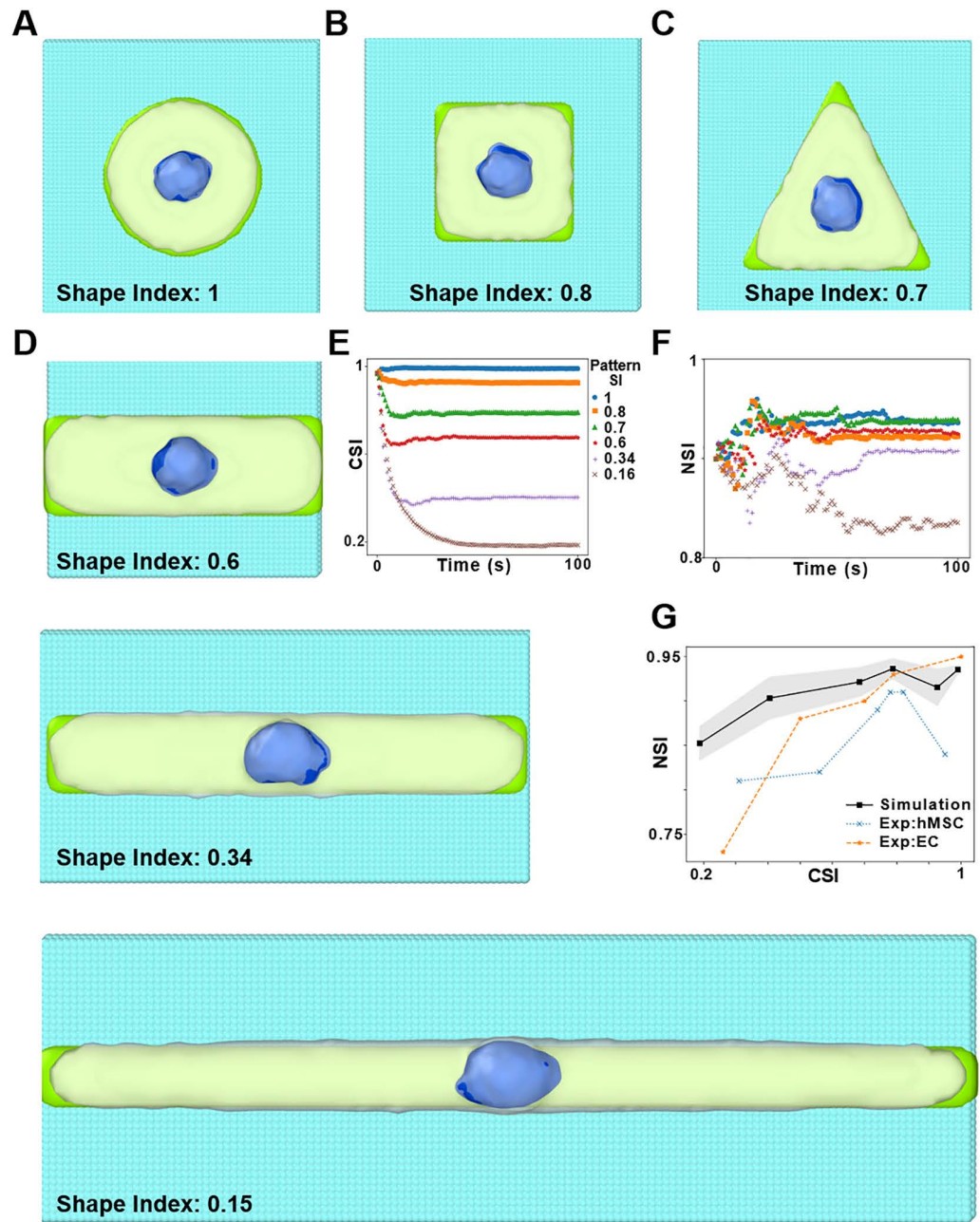

**Fig 3. Evolution of shape as cell spreads on various ECM geometries.** Simulations of cell spreading on circular **(A)**, square **(B)**, and triangular **(C)** ECM patterns, where the cell takes up the shape of the pattern. The cell membrane is depicted in white, the nucleus in blue, the ECM surface in green, and surrounding glass particles in cyan. **D)** Cell spreading on three rectangular patterns with similar area but different shape index. **E)** Cell shape index (CSI) versus time as the cell spreads on the various ECM patterns. **F)** Nuclear shape index (NSI) versus time during cell spreading on the various ECM patterns (legends of E and F are the same). **G)** Plot of NSI versus CSI for our simulations and experimental results from the literature: EC [32] and hMSC [33].

Using the SKOV3-calibrated parameter set (Fig 2) without refitting, BIOPOINT reproduced the qualitative ordering of cell spreading across ECM geometries. Cells remained most circular on circular patterns (CSI near 1; Fig 3E) and became progressively more elongated on square/triangular and then rectangular patterns (lowest CSI; Fig 3E), consistent with published micropattern experiments on EC and hMSC cells [32,33]. Across these geometries, BIOPOINT predicted limited nuclear shape changes on isotropic patterns (circle, square, triangle; Fig 3F) and modest nuclear elongation on elongated rectangles (decreased NSI; Fig 3F). When compared to published EC and hMSC datasets (Fig 3G), the model captures the qualitative coupling between cell shape (CSI) and nuclear shape (NSI) but underestimates the most extreme nuclear elongations reported for those cell types. To investigate the discrepancy, we varied the nuclear-cytoplasmic potential strength ($k_{12}$). Increasing $k_{12}$ enhanced nuclear deformability, partially improving experiment agreement (Fig 3G). However, these simulations failed to capture the most extreme nuclear elongation reported in some experimental conditions, which likely reflects missing force-transmission mechanisms (e.g., actin-mediated perinuclear compression). We therefore tested a weak nucleus–ECM interaction implemented as an attractive–repulsive Lennard-Jones potential (2–3 potential; Fig C in S1 Appendix), which can increase nuclear deformation in highly elongated patterns.

Additionally, the stresses in spreading are much smaller than those in indentation, as can be seen from the spatial distribution of stresses and spatial binning plots (Fig 4), which might non-linearly affect nuclear response. The visualization of nuclear stress in the spreading (Fig 4A) and indentation (Fig 4B) simulations show that there are considerably higher stresses in the nucleus during indentation. This can be further corroborated from the plot of average stress per particle in spatial bins along the x axis (Fig 4C). The absolute values of stresses are much more in the indented nucleus as compared to the nucleus in the spread-out cell.

These findings establish BIOPOINT's utility for modeling cell-ECM interactions, highlighting future improvements—such as incorporating cytoskeletal constraints—to replicate nuclear deformations better.

## Qualitative comparison: Nuclear deformation *during* constrained migration

Nuclear deformation is a key factor in cell migration through confined environments, influencing motility, mechanotransduction, and cell fate [11,35]. Experiments have shown that when cells traverse narrow constrictions, their nuclei experience significant shape changes and mechanical stress, which can impact migration efficiency [34,42]. Since BIOPOINT incorporates a deformable nucleus, we tested whether it can reproduce experimentally observed nuclear deformation dynamics during migration through a constriction. By simulating cell movement through a narrow passage flanked by rigid cylindrical constraints, we aimed to validate BIOPOINT's ability to replicate nuclear shape changes as cells navigate confined spaces. Furthermore, via our analysis of particle-level mechanics, we sought to provide additional mechanical information that would be inaccessible experimentally.

We designed BIOPOINT simulations based on the constricted migration experiments of Keys et al. [34] (Fig 5A). Two rigid cylindrical obstacles were placed in the simulation domain, forming a narrow constriction. As in our previous work [19], cell migration was driven by addition/removal of cytoplasmic particles at the leading/trailing edge, while nuclear positioning was controlled by a biasing force that maintains nuclear centering within the cell. During the simulation, a cell migrated through the gap, forcing its nucleus to squeeze through the confinement. Using the nuclear shape index (NSI), we tracked nuclear shape over time and observed a deformation sequence that is qualitatively comparable to the published constricted-migration dataset of Keys *et al.* [34] (MDA-MB-231; Fig 5B): the nucleus is rounded before entry ($t_1$), elongates during passage ($t_2$), and partially recovers after exit ($t_3$) (Fig 5C–5D; S5 Video). These simulations use the SKOV3-calibrated mechanical parameters without refitting; we therefore interpret this comparison as a qualitative benchmark rather than a cell-type-specific quantitative fit. BIOPOINT's particle-level representation enables spatial stress analysis across the nucleus (Fig 5E–5F), illustrating how increasing nuclear stiffness increases nuclear stresses during transit. A similar spatial strain analysis is shown in Fig D in S1 Appendix. These

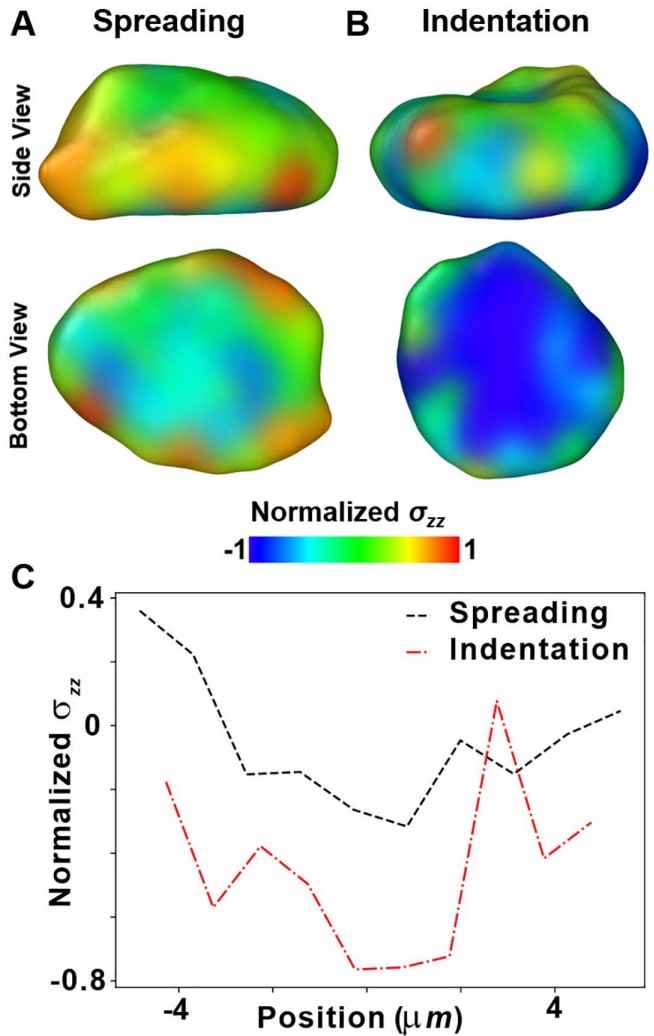

**Fig 4. Distribution of nuclear stress during indentation and spreading simulations.** Side view and the bottom surface view of A) a nucleus of a cell spread on ECM substrate and B) an indented nucleus. The nucleus has been color coded with normal component of the per-particle stress tensor ($\sigma_{zz}$). **C)** Plot of normalized $\sigma_{zz}$ versus position along one-dimensional spatial bins along the x axis.

results are consistent with experimental evidence that nuclear mechanics can influence migration efficiency in constrained environments [34].

## Discussion

BIOPOINT introduces a coarse-grained, particle-based modeling framework that explicitly represents cytoplasm, nucleus, and extracellular matrix (ECM). By modeling the nucleus as a deformable ensemble of interacting particles (Fig 1A) and incorporating an explicit ECM representation (Fig 1D), BIOPOINT enables the study of nuclear mechanics and cell-ECM interactions in ways that were impossible with previous SEM approaches. Our simulations reproduce the calibrated indentation force–time response and capture qualitative trends in cell spreading and confined migration observed experimentally [31–35]. Unlike traditional SEM, BIOPOINT incorporates mechanical parameters calibrated against experiments conducted in realistic scenarios, with cells adhered to isotropic or anisotropic ECM cues. Through single-cell indentation

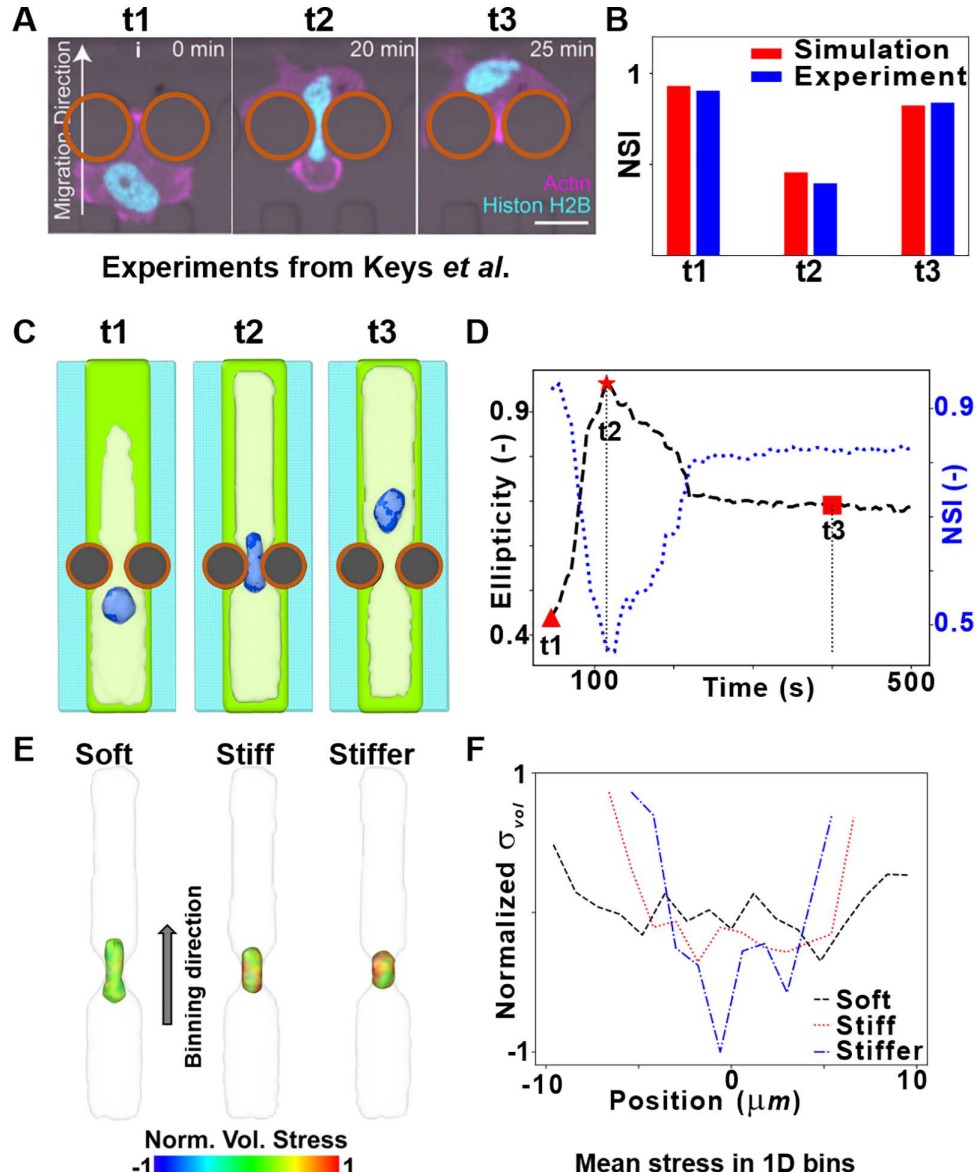

**Fig 5. Nuclear deformation when cell passes through a narrow constriction. A)** An experimental image sequence of MDA-MB-231 cells migrating through a narrow constriction, as shown by Keys *et al.* [34] (reproduced with permission). Brown circles are overlaid over the rigid constraints to enhance visualization. Scale bar: 20μm. **B)** Nuclear shape index (NSI) before (t1), during (t2), and after (t3) passing through the constriction, for our simulations and the experiment shown in **(A)**. **C)** Simulated images of a nucleus changing shape at various time points as a cell passes through a constriction. As in previous images, the ECM, cell membrane, nucleus, and glass are represented by green, white, blue, and cyan, respectively. Brown circles are overlaid over the rigid constraints to enhance visualization. **D)** Nuclear ellipticity and NSI versus time as the cell passes through the constriction. **E)** Visualization of nuclear stress distribution in nuclei with various stiffnesses. The nucleus is color-coded with volumetric stress, and the cell membrane is depicted in white. **F)** Normalized mean volumetric stress versus position in one-dimensional spatial bins along the direction of migration, indicated by the arrow in **E.**

experiments, we determined the elastic and viscous properties of both the nucleus and cytoplasm, allowing precise parameter tuning ([Fig 2]). We used Uncertainty Quantification to assess parameter sensitivity, followed by Nelder-Mead optimization, ensuring quantitative agreement between simulated and experimental force-time curves. While uncertainty

quantification was computationally efficient, running in parallel on 96-core processors (~420 simulations, ~ 1.5 hours), Nelder-Mead optimization was slower (~2 days, ~ 240 simulations). Future work could explore parallel computing with Bayesian optimization [43] to improve computational efficiency of optimization.

BIOPOINT is calibrated on a single cell-type dataset (SKOV3 indentation) and then benchmarked qualitatively against published datasets from other cell types where matched calibration datasets were not available. The qualitative agreements (e.g., Figs 3G, 5B–5D) suggest that some emergent trends are robust, but cell-type–specific quantitative predictions will require calibration and validation on matched datasets, since mechanical properties vary across cell types and conditions [44–46]. A discussion on how the parameter values are expected to change in different cell types is provided in Table B in S1 Appendix. Our cell spreading simulations confirmed that ECM geometry influences nuclear shape (Fig 3), though BIOPOINT underestimated nuclear elongation in high-aspect-ratio ECM conditions. This discrepancy likely arises from the absence of actin filament–mediated nuclear compression, which is known to contribute to nuclear deformation [32]. In addition, because BIOPOINT currently lacks an explicit actin cortex/perinuclear shell, highly flattened spreading geometries can lead to local thinning of the cytoplasmic layer around the nucleus, which can make the nucleus appear partially exposed in particle-based renders. This limitation also increases sensitivity to any added nucleus–ECM coupling (e.g., the exploratory 2–3 interaction in Fig C in S1 Appendix). Incorporating polymeric chains to model actin filaments and/or an explicit cortical/perinuclear constraint could improve BIOPOINT's ability to capture cytoskeletal contributions to nuclear reshaping [47,48]. Similarly, simulations of nuclear deformation in confined migration (Fig 5) successfully reproduced nuclear elongation under compression and partial shape recovery post-constriction, aligning with experimental data [34,35]. However, BIOPOINT does not yet explicitly model cytoskeletal filaments or microtubules, which have been shown to affect nuclear mechanics during migration. Future work could introduce active cytoskeletal components, allowing for a more detailed investigation of nuclear resilience under mechanical stress [47,48]. The number of particles used to model the nucleus could also be varied (thus varying particle size) to investigate its effect on emergent nuclear shape, although preparing a phase separated nucleus could be challenging with increase in the number of particles (See Fig G in S1 Appendix.)

BIOPOINT provides a new tool for studying nuclear mechanics, cell-ECM interactions, and constrained migration, filling a gap between continuum-based finite element models and discrete cell-based approaches such as vertex models and Cellular Potts Models. BIOPOINT extends these classical computational approaches by combining particle-based nuclear mechanics with explicit ECM modeling. Unlike FEM, which excels in stress-strain analysis but struggles with large non-linear deformations [13], BIOPOINT's particle-based approach allows emergent behavior while retaining mechanical accuracy. Compared to Cellular Potts Models, BIOPOINT explicitly represents nuclear mechanics rather than inferring nuclear deformation indirectly [15]. Finally, while vertex models effectively describe tissue-level behavior, BIOPOINT provides a more detailed subcellular perspective, making it particularly suited for studying nuclear mechanics during migration [7].

Ultimately, we believe that BIOPOINT's biophysically grounded framework, combined with experimental calibration, makes it particularly useful for biologists and bioengineers investigating emergent nuclear and cell shapes during morphogenesis. While BIOPOINT successfully models nuclear mechanics and cell-ECM interactions, further improvements could enhance its predictive power. Including cytoskeletal constraints would allow more accurate modeling of nuclear flattening under ECM confinement. Additionally, incorporating nuclear envelope rupture events could extend BIOPOINT's applicability to extreme mechanical stresses observed in confined migration. Finally, future work could explore the integration of biochemical signaling, coupling nuclear mechanics with dynamically regulated stiffness changes.

## Methods

All simulations have been performed utilizing the open-source software, LAMMPS (large-scale atomic/molecular massively parallel simulator), extended by our previously reported open-source package [19]. Similar to our previous work on

SEM² [19], visualization is via the licensed version of the software, OVITO [40]. All cells in this work are represented by 1000 particles: 900 cytoplasmic and 100 nuclear particles.

### Preparation of multi-particle nucleus

During sample preparation, initially all 1000 particles were placed randomly in a region of space. The attractive potential between nuclear particles is always stronger (Fig 1B) than the other potentials. Thus, the nuclear particles attract each other, congregate and eventually form a nucleus separated from the rest of the cell. The strength and cutoff of the nuclear potential is varied during equilibration until the nucleus is separated from the rest of the cell. The biasing force on the nucleus towards the cell center, along with the superior cytoplasmic–nuclear adhesion compared to cytoplasmic–cytoplasmic adhesion (Fig E in S1 Appendix), favors the nucleus remaining embedded within the cytoplasmic particle ensemble. In highly flattened 2D spreading simulations, local thinning of the cytoplasmic layer around the nucleus can occur; we discuss this limitation and its implications for nucleus–ECM coupling in the Discussion.

### Preparation of ECM layer

The cytoplasm, nucleus and ECM are represented by different particle types. ECM particles were placed on a lattice using the "create_atoms" command, in conjunction with the "lattice" command, in LAMMPS (input scripts provided in GitHub repository). In this simplified representation of ECM, the particles have been made static by setting their velocity to zero and there are no forces acting between them. Thus, they act like a plane of static particles, which attract the cytoplasmic particles of the cell. The attraction between cytoplasmic and ECM particles is modeled by a Lennard-Jones (12–6) potential, where σ is the zero-crossing distance, ε is the potential well-depth, and $d_c$ is the cutoff distance. The cell spreading can be tuned by adjusting the parameter, ε, of this potential (Fig 1B).

### Cell spreading

Cell spreading has been carried out by running Brownian dynamics on the cellular particles in the proximity of ECM particles, which results in the cytoplasmic particles moving towards and spreading on the ECM. The degree of cell spreading on ECM substrate depends on the parameters corresponding to the well-depth of the respective potentials -- $\varepsilon$ for the LJ potential between cytoplasm-ECM and $u_0(1-1)$ for the Morse potential between cytoplasm-cytoplasm.

### Indentation

For single cell and nuclear indentation, we first performed simulations of cell spreading on an ECM layer with area much larger than the surface area of the cell. The cell height is matched with the experimental setup of Hobson *et al*. [31]. The schematic of this experiment, shown in Fig 2A, is generated using Adobe Illustrator. The peripheral particles were immobilized so that the cell doesn't spread further when it is being indented, which is achieved experimentally due to the focal adhesions of the cell on the ECM substrate. The effect of peripheral constraint on the force-time curve can be seen in Fig H in S1 Appendix. Indentation was implemented by the "fix indent" functionality of LAMMPS with a rigid spherical indenter. During the loading period, the indenter was brought closer to the cell at a fixed rate of 1 μm/s, followed by holding the indenter still and finally retracting the indenter at the same rate. The displacement rate and loading cycle is same as that used in the experiments. The force exerted on the indenter by particles in contact with the surface is determined by the following relationship:

$$F(r) = -K(r - R)^2$$

(6)

In Eq. 6, *K* is the force constant, r is the distance between the particle and the center of the indenter, and R is the radius of the indenter. This force is only repulsive, i.e., F(r) =0 for r>R. The net force on the indenter can be extracted as an output from the LAMMPS software.

## Uncertainty Quantification

We used EasyVVUQ [49,50] with Chaospy [51] to build a 4th-order polynomial chaos expansion (PCE) for the indentation force–time output. Priors were independent Normal distributions centered at the nominal (Table 3) parameter values with $\sigma = 0.1 \cdot \mu$. Varied parameters: cytoplasm stiffness, cytoplasm viscosity, nucleus stiffness, nucleus viscosity, $\lambda$ (tuning coefficient), and $k_{12}$ (relative nuc–cyto interaction strength). From the PCE coefficients we computed moments ($\mu$, $\sigma$) and first-order Sobol indices $S_i(t)$ for the time-resolved force.

## Optimization

We compute the cost function, which is the root mean squared error (RMSE) between the transient experimental force and the simulated force curve. The parameters which have been optimized are the ones mentioned above (excluding tuning coefficient, $\lambda$). The experimental data which has been utilized has been obtained by digitizing the force-time plot of Hobson *et al*. [31] The optimization has been carried out with the Nelder-Mead algorithm from SciPy [52]. The initial values (Table 3) were chosen by analyzing the samples used in the UQ and the impact of individual parameters based on the force curve. In this way, the initial guess of the force curve already matched relatively well with the experimental curve. For convergence, the default tolerance of x values (xatol) and function values (fatol) have been considered, which is 1e-4 for both. The final optimized values of parameters are provided in Table 3. Details of our optimization method and code can be found in our GitHub page.

## Cell spreading simulations on ECM patterns

For the simulations of cell spreading on ECM patterns, ECM particles were created on a lattice with different planar shapes such as circle, square, triangle and rectangle. Outside the ECM patterns, inert particles have been created to mimic glass substrate of in vitro experiments. The functional form of the potential between glass and cytoplasmic particles is the same as Eq. 5 but the cutoff distance is the zero-crossing distance, i.e., there is only repulsive potential in order to account for volume exclusion. On performing Brownian dynamics on the cellular particles in the proximity of ECM particles, the cellular particles gradually take up the shape of the ECM pattern as shown in Fig 3. We chose 100/900 nuclear/cytoplasmic particles in order to have the minimum number of nuclear particles to reproduce emergent nuclear shapes [16], but conducted additional simulations with 200 nuclear particles representing the nucleus (see Fig F in S1 Appendix).

## Cellular and nuclear shape analysis

The projection of the particle coordinates in a two-dimensional plane of the ECM particles is considered. A convex hull is generated over these 2D coordinates by using the algorithm provided by SciPy. The perimeter ($P$) and surface area ($A$) of the convex hull is then computed. Finally, the cell shape index (CSI) is then determined using Eq. 7:

**Table 3. Parameter values in UQ and Optimization.**

| Parameter | Mean used in UQ | Optimization: Initial Value | Optimization: Final Value |
|---|---|---|---|
| $\kappa_0$ | $1.5 \times 10^{-2}$ N m$^{-1}$ | $1.2 \times 10^{-2}$ N m$^{-1}$ | $1.208 \times 10^{-2}$ N m$^{-1}$ |
| $\eta_0$ | $5 \times 10^{-2}$ N s m$^{-1}$ | $5 \times 10^{-2}$ N s m$^{-1}$ | $4.975 \times 10^{-2}$ N s m$^{-1}$ |
| $\kappa_{nuc}$ | $6 \times 10^{-2}$ N m$^{-1}$ | $4.5 \times 10^{-2}$ N m$^{-1}$ | $4.531 \times 10^{-2}$ N m$^{-1}$ |
| $\eta_{nuc}$ | $2.2 \times 10^{-1}$ N s m$^{-1}$ | $2 \times 10^{-1}$ N s m$^{-1}$ | $2.054 \times 10^{-1}$ N s m$^{-1}$ |
| $k_{12}$ | 1.5 | 1.5 | 1.51 |
| $\lambda$ | 0.75 | -- | -- |

$$CSI = \frac{4\pi A}{P^2}$$

(7)

Following a similar procedure, the nuclear shape index (NSI) is computed. For comparison with experiments, we have also plotted values of CSI versus NSI of a couple of cell lines. For extremely tapering shapes, such as the experimental cell shape inside the constriction, we use alpha shape method instead of convex hull method as it gives more accurate estimation of nuclear shape.

### Cell migration through constriction

For the simulations involving a cell passing through cylindrical obstructions, we carried out migration via the addition and removal of particles on a rectangular ECM pattern (Input script available on GitHub). We leveraged the simulation software SEM++ (Milde, 2014, [18]), where the particle ensembles move as we add a particle on one side of the cell (leading edge) and remove a particle from the other (trailing edge). This process reflects the polymerization/depolymerization of the actin filaments which take place during cell migration at the leading/trailing edge. The force balance described in Eq. 1 generates rapid re-adjustments of particle positions and velocities in the ensemble which restore cell rheology. A biasing force on the nucleus towards the cell center on a two-dimensional plane repositions it throughout the migration simulation [19]. The cylindrical obstructions have been implemented by rigid indenters with "fix indent" functionality of LAMMPS.

The NSI is computed by following the procedure mentioned above (similar to CSI). The NSI of the experimental images is measured by digitizing the shape of the nucleus and generating a convex hull over the coordinates.

### Nuclear ellipticity computation

For nuclear shape analysis, a three-dimensional ellipse has been fitted over the coordinates of nuclear particles using an existing Python script [53]. The ellipticity of the fitted ellipse, $f$, is measured using the relation:

$$f = \frac{a - b'}{a} \quad , \quad b' = \frac{b + c}{2}$$

(8)

In Eq. 8, $a$ is the largest radius of the fitted ellipse and $b$ and $c$ are the two smaller radii.

### Visualization, stress and strain computations

Generating a surface mesh over the particles has been carried out in OVITO [40], employing a Gaussian density method. The per-particle stress can be computed utilizing the "compute stress/atom" functionality of LAMMPS. By transferring the particle-level stress on the enclosing surface in OVITO, we could visualize the stress distribution in the nuclear surface mesh. The average volumetric stress per particle in 1D spatial bins then provided us with a measure of the nuclear stress distribution along the direction of binning. Additionally, the per-particle shear strain of the nuclear particles with respect to the initial configuration was computed via the "Atomic Strain" modifier of OVITO [40].

### Particle stress

The per-particle stress has been determined utilizing the "compute stress/atom" functionality of LAMMPS. This yields per-particle virial stress, which has the units of energy and can be converted to stress units if divided by the particle volume. We compute the mean volumetric stress of each particle as, $\sigma_m = \frac{\sigma_{xx} + \sigma_{yy} + \sigma_{zz}}{3}$. In order to distinctly visualize the distribution of stresses across the subcellular material, we have normalized the stresses with maximum and minimum values of a range within which stresses of the majority of particles lie. In order to project particle stress onto the surface, we used OVITO [40], where existing attributes of the input particles located at the surface are copied over to the vertices of the constructed surface mesh."

## Supporting information

**S1 Appendix.** Fig A. Quantification of spreading by measuring projected surface area of cells. Fig B. Spatial stress distribution in the bottom surfaces of cells on patterns. Fig C: Drastic nuclear shape change in rectangular patterns. Fig D: Spatial strain analyses for cell migration through a constriction. Fig E: Potential between cytoplasmic and nuclear particles. Fig F: Evolution of shape in the cell with 20% nucleus -- 200/800 nuclear/cytoplasmic particles. Fig G: Force versus time for single cell indentation simulations with different values of $N_p$. Fig H: Effect of immobilization of peripheral particles. Table A. Values of the LJ parameters and the corresponding cell height in cell spreading experiments. Table B. BIOPOINT rheological parameters and expected variation across cell types, with representative experimental ranges.
(PDF)

**S1 Video. AFM indentation of a cell with nucleus.**
(MP4)

**S2 Video. Nuclear stress during indentation.**
(MP4)

**S3 Video. Nuclear strain during indentation.**
(MP4)

**S4 Video. Cell spreading on patterns.**
(MP4)

**S5 Video. Cell migrating through a constriction.**
(MP4)

**S6 Video. Nuclear stress while passing through a constriction.**
(MP4)

## Acknowledgments

We would like to thank Constanze Kalcher and Alexander Stukowski of OVITO GmbH for the discussions which benefitted several of our post-processing tasks, including quantifying the number of cytoplasmic particles inside the surface mesh encompassing nuclear particles.

## Author contributions

**Conceptualization:** Sandipan Chattaraj, Francesco Pasqualini.

**Data curation:** Sandipan Chattaraj, Julius Zimmermann, Francesco Pasqualini.

**Formal analysis:** Sandipan Chattaraj, Julius Zimmermann, Francesco Pasqualini.

**Funding acquisition:** Francesco Pasqualini.

**Investigation:** Sandipan Chattaraj, Julius Zimmermann, Francesco Pasqualini.

**Methodology:** Sandipan Chattaraj, Julius Zimmermann.

**Project administration:** Francesco Pasqualini.

**Resources:** Francesco Pasqualini.

**Software:** Francesco Pasqualini.

**Supervision:** Francesco Pasqualini.

**Validation:** Sandipan Chattaraj, Julius Zimmermann.

**Visualization:** Sandipan Chattaraj, Julius Zimmermann.

**Writing – original draft:** Sandipan Chattaraj, Francesco Pasqualini.

**Writing – review & editing:** Sandipan Chattaraj, Julius Zimmermann, Francesco Pasqualini.

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
