## [Decision Letter · Decision Letter 0]

14 Aug 2025

BIOPOINT: A particle-based model for probing nuclear mechanics and cell-ECM interactions via experimentally derived parameters.

PLOS Computational Biology

Dear Dr. Pasqualini,

Thank you for submitting your manuscript to PLOS Computational Biology. After careful consideration, we feel that it has merit but does not fully meet PLOS Computational Biology's publication criteria as it currently stands. Therefore, we invite you to submit a revised version of the manuscript that addresses the points raised during the review process.

Please submit your revised manuscript within 60 days Oct 14 2025 11:59PM. If you will need more time than this to complete your revisions, please reply to this message or contact the journal office at ploscompbiol@plos.org. Please include the following items when submitting your revised manuscript:

We look forward to receiving your revised manuscript.

Kind regards,

Christopher E Miles

Academic Editor

PLOS Computational Biology

Dimitrios Vavylonis

Section Editor

PLOS Computational Biology

**Additional Editor Comments:**

First, I would like to apologize for the delay in handling this manuscript. We had an exceptionally difficult time securing reviewers, likely due to the time of year.

Ultimately, I agree with the reviewer concerns about the manuscript's clarity, especially regarding parameter definitions and model presentation. I also like to raise a concern about the framework's novelty. It remains unclear what BIOPOINT provides that hasn't already been addressed by previous continuum-based methods. For instance, Estabrook et al. (2021), who reconstruct nuclear deformation forces from experimental images using elastic solid and shell models, and Maxian et al. (2020), who modeled cell–ECM–nucleus mechanics including hydrodynamics in a FEM setting. Without a clear advantage over these established approaches, the novelty of BIOPOINT appears limited to a different computational implementation rather than a conceptual advance. If the authors believe there is indeed a novel advantage to this framework that is otherwise unobtainable through these other methods, and can convincingly rewrite the manuscript to convey and demonstrate this, the manuscript may be considered for publication. However, until then, I cannot recommend it for such.

Estabrook ID, Thiam HR, Piel M, Hawkins RJ (2021). Calculation of the force field required for nucleus deformation during cell migration through constrictions. PLoS Computational Biology 17(5): e1008592. https://doi.org/10.1371/journal.pcbi.1008592

Maxian O, et al. (2020). Computational estimates of mechanical constraints on cell migration through dense extracellular matrices. PLoS Computational Biology 16(10): e1008160. https://doi.org/10.1371/journal.pcbi.1008160

**Journal Requirements:**

3) Please ensure that all Table files have corresponding citations within the manuscript. Currently, Table 1 in your submission file inventory does not have an in-text citation. Please include the in-text citation of the table.

Potential Copyright Issues:

i) Figures 1A, 1D, and S1A. Please confirm whether you drew the images / clip-art within the figure panels by hand. If you did not draw the images, please provide (a) a link to the source of the images or icons and their license / terms of use; or (b) written permission from the copyright holder to publish the images or icons under our CC BY 4.0 license. Alternatively, you may replace the images with open source alternatives. See these open source resources you may use to replace images / clip-art:

**Reviewers' comments:**

Reviewer's Responses to Questions

Reviewer #1: The authors present BIOPOINT, a subcellular-element model (SEM) type cell based modelling framework based on the molecular dynamics software LAMMPS. BIOPOINT is a new iteration of previous SEM software SEM2 and SEM++. In this manuscript, the software has been extended to represent the cell nucleus with multiple particles and a two-dimensional representation of the extracellular matrix has also been implemented. Additionally, the authors calibrate the parameters of their model with previously-published mechanobiological experiments including atomic force microscopy microindentation and cell migration in confinement.

Development and experimental validation of better modelling software is important for the computational biology community, for which this manuscript has value. The authors also demonstrate that BIOPOINT can be used to determine internal stresses and strain in the cytosol and in the nucleus, a novel mechanobiological model prediction possible with this model that could be used to explore nuclear mechanics and chromatin organization.

I have some questions and comments to the authors.

1) The abstract states that "traditional SEM frameworks lack explicit representation of nuclear mechanics and cell-extracellular matrix (ECM) interactions".

There are previous SEM works that simulate tissue mechanics that explicitly represent a multi-particle nucleus and extracellular matrix (for a recent example see Kumar et al. npj Systems Biology and Applications 10.1 (2024): 49.). Hence this statement in the abstract is not correct and should be revised.

2) Related to the previous point, the introduction should provide better contextualization of prior work. The authors briefly mention and dismiss other modelling frameworks such as vertex and cellular Potts models. This is disappointing, as each framework has its merits and has been used to successfully simulate mechanobiology. In the authors' own previous work (Chattaraj et al. APL bioengineering 7.4 (2023).), they do a much better job at describing the broader context of cell based models including citing relevant reviews (Metzcar et al. JCO clinical cancer informatics 2 (2019): 1-13.) and previous relevant SEM work (e.g. Nematbakhsh et al. PLoS computational biology 16.8 (2020): e1008105.).

In addition, some other relevant papers the authors should acknowledge include work parametrizing cell based models to tissue mechanics (Pathmanathan et al. Physical biology 6.3 (2009): 036001.) and work modelling nuclear deformation during cell migration (Scianna and Preziosi. Axioms 10.1 (2021): 32.).

3) The authors should provide a better description of what their model entails, which is lacking. I realize there is a prior publication that this model builds on, but at least a high-level summary would be needed.

I notice also that equation numbering starts at 4 in the methods. Are equations 1-3 missing from the text?

4) The authors rigorously calibrate their model to experimental results, which is commendable. However, different cell types are used for calibration. Realistically, mechanical properties differ between cell types and could potentially vary over time, e.g. as a cell switches from stationary to motile states. The authors should comment on this in the discussion.

5) What interaction potential did the authors choose for interactions between nucleus and cytoplasm particles?

How does the model ensure that the nucleus particles stay surrounded by cytoplasm particles?

6) Regarding Figure 3G: The line connecting the dots for "Exp:hMSC" does not look correct. It seems to move backwards along the x-axis.

Something similar occurs in Figure S3A in the supplementary material.

7) The nuclear deformation in the experimental images of Figure 4A looks very extreme compared to the simulation in Figure 4C. Yet the bars in the quantification in Figure 4B show the opposite: The simulation appears to have a more extreme NSI than the experiment. Can the authors explain why?

8) Did I understand right that all cytoplasmic particles can adhere to the ECM and thus there is not a special "membrane" particle? Does this mean that the extent of cell spreading is mediated by the balance between cytoplasm-ECM adhesion and cytoplasm-cytoplasm adhesion? Or will all cells eventually maximally spread once equilibrium is reached?

9) What is the lowest number of nuclear or cytosolic particles that can still reproduce the experimental data? This would be interesting to know, given that computational cost scales with number of particles.

In a similar vein, why did the authors choose 100 nucleus particles?

10) For Figure 2C and 2D, the authors do an uncertainty quantification (UQ). I had some trouble understanding what these panels show, and would appreciate more explanation.

I understand from the methods that the authors varied parameter values and repeated the simulation using a Python package. Is 2C showing summary statistics of predictions assuming the parameters vary around a given starting value? How much do the starting values affect the outcome of the UQ? Is this what 2D is showing?

11) The authors show some interesting data in the supplementary material. For example Figure S4 is quite nice, and in this reviewer's opinion should be showcased in the main manuscript, perhaps in Figure 2.

12) About supplementary Figure S2, where the authors show the spatial stress distribution at the bottom surface of a cell adherent to the ECM. Why is there a tensile/positive stress in the boundary and a compressive/negative stress within the cell? Do the authors have an explanation of how this stress distribution emerges?

13) For Figure 2, the authors state: "A BIOPOINT cell was spread on an ECM layer, with the nucleus positioned near the top surface." From the wording it's unclear if this behavior is an emergent property of the simulation or if the authors encoded this configuration.

14) The authors state that BIOPOINT "optimizes cell mechanics" (last paragraph before Figure 2 is shown). What do the authors mean by "optimize"? Perhaps they could reword for clarity.

15) About the parameter uncertainty quantification: To what extent are the values of the parameters uniquely identifiable?

16) Regarding Figure 3F: It would improve figure legibility if the legend was moved outside the plot area and if the points were connected by lines.

Reviewer #2: The authors have presented a method of performing simulations of single cells with a particle representation for both the nucleus and the cytoplasm. The advance here is that while in previous work, the nucleus was represented by a single rigid particle, here, the nucleus is represented by a number of smaller particles that experience a potential interaction with each other. The expectation is that nuclear particles, experiencing certain potential-based interactions with each other, will naturally cluster and phase separate from the cytoplasmic particles. Hence the model is expected to behave similarly to a 2-phase model.

The model presented here depends on multiple parameters including single particle stiffness parameters and particle-to-particle interaction strengths. The utility of the model is that once these parameters are trained on data, they can be used on future numerical experiments for the same cell type. I believe this method could be useful to reserachers that have access to single cell experiments and are therefore able to calibrate the model. After that, the calibrated model could be used as a predictive tool for future experiments on the same cell type.

The strength of the manuscript is that researchers may indeed find the model useful and implementable in their own labs. The weakness of the manuscript is that although it presents a useful tool, it does not give insight to biological phenomena or biophysics. The majority of the comments I have about the paper is regarding its clarity when it comes to explaining and presenting the model, as well as asking for more complete justification to the modeling assumptions and methods. The points with ** are likely the more important points to address.

1. I want to stress (from what I understand) that this model may be limited to the use case I mentioned earlier, i.e. it is a numerical model that requires calibration before use; and future applications of the model must be on the same cell type and similar experimental conditions as when it was calibrated. The authors should mention this use case and limitation.

2. The paper would benefit in general from the Methods section being merged with the main text, or for some portion of the Methods to be included in the main text. For example, the parameter \kappa_nuc and \kappa_0 appear without explanation in the main text other than to say they are a "stiffness potential". Equations like (4) and (5) could be moved to the main text. The parameter tables should also be moved to the main text. The paper is about your method, so all the parameters should appear in the main part of the paper.

3. The "stiffness potentials" all need to be explained with a functional form or physical descriptor (is it Young's modulus? Is it elastic?), something beyond saying it is parameter found in LAMMPS.

4**. Similar to 3, all parameters all need to be defined and explained in the main text. Even in the Methods section, explanations are missing. For example, in the text prior to Eq. (6), the parameters are mentioned only by name without explanation. What is tuning coefficient? What is u_0 in Eq. (6)? Eq. (6) itself is not clear, what is u_0(1-2)?

4b**. Viscosity is not explained in the main text, how is viscosity simulated between particles? Viscosity, scaling factor, shifting factor, etc all appear in the Methods, they all need explanation.

5. How the per-particle stress is determined needs to be explained (other than it comes from LAMMPS). The explanation of how the stress is projected onto the nuclear surface should explained better.

6**. How is the membrane defined and evolved?

7**. The fit or calibration of the adhesion potential between ECM particles and cytoplasmic particles are never presented (or I just haven't found it). There are 3 parameters associated with Lennard-Jones, how were these 3 parameters fitted in your paper? Their values are also not given in the Methods.

8**. Equation 5 in the Methods give how the force for the indentation experiment is defined. Is the force between the indentation particle and every particle in the simulation? Or is the force only between the indentation particle and the immediate particles it is in contact with and 0 otherwise? If it is the former, that is not very physical and needs to be justified. The latter choise is the better implementation and the force should be given in the main text.

9**. In the Cell migration through constriction section, why is the migration simulated via the addition and removal of particles? The particles are lagrangian particles, so isn't it more sensible to simulate migration by a force exerted between cytoplasmic and ECM particles, or an extra velocity added to the cytoplasmic particles (which is not a great modeling choice, but still better than deleting and adding particles). How is the adding/deleting of particles justified?

10**. How are the sizes and total number of the particles chosen? Should it be according to the volume of the cell you are modeling? For example, if we were to use 1200 instead of 1000 particles to start, how would the parameters change? In other words, does using 1200 particles instead of 1000 require recalibration or can the parameters be mapped to new values?

11. You mention that around 8% of cytoplasmic particles end up in the nucleus, that doesn't seem to be negligeable, even though it is not very large, can you comment on whether this is worrying and whether the 8% can be reduced.

12**. Are the experiments you fit in the indentation experiment the same cell type as the ones you use later to fit the cell spreading experiments?

13. Related to 12 and 10, could you mention what range of values for the parameters could be expected if the cell types were to change?

14. You say "the projected area expanding proportionally to the adhesion potential", I don't think you show proportionality, you show that one increases as the other increases, which is not necessarily a proportional relationship.

15**. Are the immobilized peripheral cytoplasmic particles used in the indentation experiment necessary? Can't the interaction between cytoplasmic particles and the ECM and/or the attraction between cytoplasmic particles themselves keep the cell from spreading too much? Please justify the immobile particles.

16. As a matter of style, the plot in Figure 4F should just be a line plot without the filling. It currently looks unnecessarily complicated.

17. Could the discrepancy between the experiment and your nuclear shape (Figure 3G) be due to using a nucleus in your model that's too small? It looks like the nucleus doesn't need to deform much to fit into the ECM pattern, if the nucleus was larger, it might deform more.

**Have the authors made all data and (if applicable) computational code underlying the findings in their manuscript fully available?**

The PLOS Data policy requires authors to make all data and code underlying the findings described in their manuscript fully available without restriction, with rare exception (please refer to the Data Availability Statement in the manuscript PDF file). The data and code should be provided as part of the manuscript or its supporting information, or deposited to a public repository. For example, in addition to summary statistics, the data points behind means, medians and variance measures should be available. If there are restrictions on publicly sharing data or code —e.g. participant privacy or use of data from a third party—those must be specified.requires authors to make all data and code underlying the findings described in their manuscript fully available without restriction, with rare exception (please refer to the Data Availability Statement in the manuscript PDF file). The data and code should be provided as part of the manuscript or its supporting information, or deposited to a public repository. For example, in addition to summary statistics, the data points behind means, medians and variance measures should be available. If there are restrictions on publicly sharing data or code —e.g. participant privacy or use of data from a third party—those must be specified.requires authors to make all data and code underlying the findings described in their manuscript fully available without restriction, with rare exception (please refer to the Data Availability Statement in the manuscript PDF file). The data and code should be provided as part of the manuscript or its supporting information, or deposited to a public repository. For example, in addition to summary statistics, the data points behind means, medians and variance measures should be available. If there are restrictions on publicly sharing data or code —e.g. participant privacy or use of data from a third party—those must be specified.requires authors to make all data and code underlying the findings described in their manuscript fully available without restriction, with rare exception (please refer to the Data Availability Statement in the manuscript PDF file). The data and code should be provided as part of the manuscript or its supporting information, or deposited to a public repository. For example, in addition to summary statistics, the data points behind means, medians and variance measures should be available. If there are restrictions on publicly sharing data or code —e.g. participant privacy or use of data from a third party—those must be specified.

Reviewer #1: Yes

Reviewer #2: Yes

PLOS authors have the option to publish the peer review history of their article (what does this mean?). If published, this will include your full peer review and any attached files.). If published, this will include your full peer review and any attached files.). If published, this will include your full peer review and any attached files.). If published, this will include your full peer review and any attached files.

...

Reviewer #1: No

Reviewer #2: No

**Figure resubmission:**

**Reproducibility:**



---

## [Decision Letter · Decision Letter 1]

22 Jan 2026

PCOMPBIOL-D-25-00853R1

BIOPOINT: A particle-based model for probing nuclear mechanics and cell-ECM interactions via experimentally derived parameters.

PLOS Computational Biology

Dear Dr. Pasqualini,

Thank you for submitting your manuscript to PLOS Computational Biology. After careful consideration, we feel that it has merit but does not fully meet PLOS Computational Biology's publication criteria as it currently stands. Therefore, we invite you to submit a revised version of the manuscript that addresses the points raised during the review process.

We look forward to receiving your revised manuscript.

Kind regards,

Christopher E Miles

Academic Editor

PLOS Computational Biology

Dimitrios Vavylonis

Section Editor

PLOS Computational Biology

**Additional Editor Comments:**

While the reviewers are generally positive about the improvements, Reviewer 2 has identified critical instances where the interpretation of the results remains overclaimed, specifically regarding the validation against different cell types and the sensitivity of particle numbers.

**Journal Requirements:**

1) Please amend your detailed Financial Disclosure statement. This is published with the article. It must therefore be completed in full sentences and contain the exact wording you wish to be published.

**Reviewers' comments:**

Reviewer's Responses to Questions

**Comments to the Authors:**

Reviewer #1: In the revised manuscript, the authors addressed most concerns well. I have a few remaining comments:

1) In the revised introduction, there is a long section starting with "Prior SEM implementations have typically represented the nucleus..." and ending in "...influences cell spreading and mechanosensing". This section should include references to these prior SEM implementations.

2) The authors write "cell mechanics is lost (e.g., Potts models [22])", implying that the cellular Potts model does not represent cell mechanics. First, the Potts model is not the same as the cellular Potts model, so please adjust the wording. Second, the statement is factually incorrect. Cellular Potts models do represent cell mechanics at the cell membrane as the balance of forces between cell pressure, surface tension due to actin cortex contractility, and interfacial adhesion (Magno et al. 2015). Energy is related to force via integration, and interconversions within the cellular Potts formalism were shown by Rens & Edelstein-Keshet (2019). Similar energy-based formulations are also used in particle-based models (including molecular dynamics-based methods) and vertex models (Fletcher et al. 2014), which the authors agree can represent aspects of cell mechanics. I suggest rewording to clarify that internal or cytosolic cell mechanics are not represented, which is true for both cellular Potts models and vertex based models.

References:

Magno, R., Grieneisen, V.A. & Marée, A.F. The biophysical nature of cells: potential cell behaviours revealed by analytical and computational studies of cell surface mechanics. BMC Biophys 8, 8 (2015). https://doi.org/10.1186/s13628-015-0022-x

Rens EG, Edelstein-Keshet L (2019) From energy to cellular forces in the Cellular Potts Model: An algorithmic approach. PLOS Computational Biology 15(12): e1007459. https://doi.org/10.1371/journal.pcbi.1007459

Fletcher, Alexander G., et al. "Vertex models of epithelial morphogenesis." Biophysical journal 106.11 (2014): 2291-2304. https://doi.org/10.1016/j.bpj.2013.11.4498

3) In the caption of Figure 1 F, the authors mention "The cell membrane is white". Please reword to clarify that there is not an actual cell membrane, but the renders show the peripheral particles as white (or something along these lines -- avoid confusion about there being any membrane particles).

4) How often do you observe that some of the nucleus particles are exposed to the medium without any peripheral cytosolic particles surrounding them? At least in the OVITO renders and supplementary movies, this seems to occur fairly often. Is this a source for concern? Could this explain why adding a 2-3 potential between nucleus and ECM leads to greater nuclear deformation? Should this be addressed in future studies (-> discussion material)?

5) Is the 2-3 potential you test in Figure S3 purely repulsive? Or what form does it have? Perhaps it would be good to show it in a sub-panel in the figure.

Reviewer #2: The manuscript is now written with much higher clarity in terms of the Methods. Most of my questions have been addressed. However, I *strongly suggest* the following 2 points be addressed in the Results sections before acceptance for publication.

This is to make the results clearer, more accurate, and more useful for the Plos Comp Bio audience.

1. This is regarding my original comment 2.10 which discusses the number of particles used in the simulation. It is not clear to me that going from N_p=1000 to N_p=1200 constitutes a small change in the force profile, or that N_p=1200 "matches well" with N_p=1000. In fact, with N_p=1000, it seems that force decreases during the holding time in the indentation experiment, while with N_p=1200 and N_p=10000, the force is almost constant during the holding time (even though the value of constant force between N_p=1200 and N_p=10000 is quite different during the holding time).

Without a full analysis of how various N_p would generate different values for the other parameters after the data is recalibrated, and an analysis of the range of the other parameter values that would result form changing N_p, you should only claim that N_p must be set from the beginning and that changing N_p requires recalibration. You should *not* attempt to say anything quantitative like: "for small variations in the number of particles, we could still continue with the same parameter values, but for a drastic change...", as your analysis is not able to fully show this.

2. This is regarding my original comment 2.12 in which I asked about the the cell types you used to fit the numerical SEM experiments. You clarified in your response that the calibration was done only on the indentation experiments with SKOV3 and that the inferred parameters from the indentation experiments were then used for the spreading and migration simulations. You also mentioned that the later comparisons were with other cell types even though the model was calibrated with SKOV3.

This needs to be made clear in the sections "Validation of BIOPOINT via cell spreading on ECM patterns" and "Validation of BIOPOINT in nuldear Deformation during constrained migration", and not just in the "Discussion" section. As the manuscript currently stands, the Results sections "Validation of BIOPOINT..." are hard to follow simply because you do not state clearly that you perform simulations using parameters calibrated on SKOV3 cell-indentation experiments only. You need to mention this, and you need to mention very explicitly that you now compare these new simulations (spreading, migration) to experiments in *other* cell types (EC and hMSC), so that the experimental data and the simulations are not expected to match, as the calibration was done with SKOV3. The data and simulations are only expected to have some qualitative correspondence. This sets up the correct expectation for the reader.

Moreover, in these sections, you should get rid of language like "successfully recapitulated", "matched experiments", "matched experimental results", and "consistent with experimental observations"; as the cell types are different, you cannot "recapitulate" or "match" anything, and this language is confusing. You should mention only qualitative agreement, things trending in the same direction, etc, as quantitative agreement is *not* expected. You should also mention the cell types EC and hMSC in the *text* of "Validation of BIOPOINT via cell spreading" and not just in the figure legends. If possible, you can incorporate your discussion of how parameters can differ between different cell types and how much difference you think exist between SKOV3 (which you calibrated on) versus EC and hMSC (which you compare to the other simulations).

Other suggestions:

1. You have not specified how u_0(1-2) and u_0(1-1) in Eq. (4) generalize from Eq. (3d), for example, to get u_0(1-2), do you need to replace 2R_cell with R1+R2?

2. The colors of the graphs in Fig. 2D, 5F are hard to tell apart. It's hard to see the difference between blue and green.

3. Fig 4C does not need the filling in the graph.

**Have the authors made all data and (if applicable) computational code underlying the findings in their manuscript fully available?**

The PLOS Data policy requires authors to make all data and code underlying the findings described in their manuscript fully available without restriction, with rare exception (please refer to the Data Availability Statement in the manuscript PDF file). The data and code should be provided as part of the manuscript or its supporting information, or deposited to a public repository. For example, in addition to summary statistics, the data points behind means, medians and variance measures should be available. If there are restrictions on publicly sharing data or code —e.g. participant privacy or use of data from a third party—those must be specified.requires authors to make all data and code underlying the findings described in their manuscript fully available without restriction, with rare exception (please refer to the Data Availability Statement in the manuscript PDF file). The data and code should be provided as part of the manuscript or its supporting information, or deposited to a public repository. For example, in addition to summary statistics, the data points behind means, medians and variance measures should be available. If there are restrictions on publicly sharing data or code —e.g. participant privacy or use of data from a third party—those must be specified.requires authors to make all data and code underlying the findings described in their manuscript fully available without restriction, with rare exception (please refer to the Data Availability Statement in the manuscript PDF file). The data and code should be provided as part of the manuscript or its supporting information, or deposited to a public repository. For example, in addition to summary statistics, the data points behind means, medians and variance measures should be available. If there are restrictions on publicly sharing data or code —e.g. participant privacy or use of data from a third party—those must be specified.requires authors to make all data and code underlying the findings described in their manuscript fully available without restriction, with rare exception (please refer to the Data Availability Statement in the manuscript PDF file). The data and code should be provided as part of the manuscript or its supporting information, or deposited to a public repository. For example, in addition to summary statistics, the data points behind means, medians and variance measures should be available. If there are restrictions on publicly sharing data or code —e.g. participant privacy or use of data from a third party—those must be specified.

Reviewer #1: Yes

Reviewer #2: None

PLOS authors have the option to publish the peer review history of their article (what does this mean?). If published, this will include your full peer review and any attached files.). If published, this will include your full peer review and any attached files.). If published, this will include your full peer review and any attached files.). If published, this will include your full peer review and any attached files.

...

Reviewer #1: No

Reviewer #2: No

**Figure resubmission:**
---

## [Editor Report · Decision Letter 2]

9 Mar 2026

Dear Prof. Pasqualini,

We are pleased to inform you that your manuscript 'BIOPOINT: A particle-based model for probing nuclear mechanics and cell-ECM interactions via experimentally derived parameters.' has been provisionally accepted for publication in PLOS Computational Biology.

Best regards,

Christopher E Miles

Academic Editor

PLOS Computational Biology

Dimitrios Vavylonis

Section Editor

PLOS Computational Biology

---

## [Editor Report · Acceptance letter]

PCOMPBIOL-D-25-00853R2

BIOPOINT: A particle-based model for probing nuclear mechanics and cell-ECM interactions via experimentally derived parameters.

Dear Dr Pasqualini,

I am pleased to inform you that your manuscript has been formally accepted for publication in PLOS Computational Biology. Your manuscript is now with our production department and you will be notified of the publication date in due course.

With kind regards,

Judit Kozma
